# Full Dilatation Caesarean Section and the Risk of Preterm Delivery in a Subsequent Pregnancy: A Historical Cohort Study

**DOI:** 10.3390/jcm9123998

**Published:** 2020-12-10

**Authors:** Lauren Jade Ewington, Siobhan Quenby

**Affiliations:** 1Division of Biomedical Sciences, University of Warwick, Coventry CV2 2DX, UK; S.Quenby@warwick.ac.uk; 2Department of Women and Children, University Hospitals Coventry and Warwickshire, Coventry CV2 2DX, UK

**Keywords:** caesarean section, premature birth, labour stage, second

## Abstract

Full dilatation caesarean sections (CS) have increased risk of uterine extensions, which leads to cervical trauma that has been associated with an increased risk of spontaneous preterm birth (sPTB) in a subsequent pregnancy. The aim of this study was to determine if CS at full dilatation increased the risk of sPTB in a subsequent pregnancy in our unit. A historical cohort study was performed on women delivered by emergency CS between 2008–2015 (*n* = 5808) in a university hospital who had a subsequent pregnancy in this time frame (*n* = 1557). Women were classified into two exposure groups; those who were 6–9 cm and those fully dilated at index CS. The reference group was CS at 0–5 cm dilated. The primary outcome was sPTB < 37 weeks’ gestation. CS at 6–9 cm or fully dilated did not significantly increase the odds of sPTB in a subsequent pregnancy (aOR 1.64, 95% CI: 0.83–3.28, *p =* 0.158; aOR 1.86, 95% CI: 0.91–3.83; *p =* 0.090, respectively). However, a short interpregnancy interval of <1 year significantly increased the odds of sPTB in a subsequent pregnancy (aOR 3.10, 95% CI: 1.71–5.61). This study has found a short interpregnancy interval following a CS conferred a higher risk of sPTB than full dilatation CS. This finding highlights postnatal contraception and increased surveillance of women with short interpregnancy interval post CS as possible interventions to reduce sPTB.

## 1. Introduction

In 2017–2018, 168,946 (28.4%) women in the UK were delivered by caesarean section (CS) [1]. The CS rate in the UK is increasing, with a corresponding increase in the rate of CS in the second stage of labour [2,3]. CS at full cervical dilatation can lead to significant perinatal and maternal morbidity. Maternal morbidity arises from an increased risk of interoperative trauma [4]. Approximately, 24% of full dilatation CSs will sustain an extension to the uterine incision, further increasing the risk of maternal morbidity [5].

It is estimated that 15 million neonates are born before 37 weeks’ gestation globally per annum [6]. The aetiology of preterm birth (PTB) is complex and multifactorial; however, cervical weakness is a known risk factor in its pathogenesis. An extension to the uterine incision at full dilatation CS into the cervix means that the cervix then undergoes structural change and remodelling as part of the healing process. This cervical damage and repair could render the cervix weaker, so that it dilates prematurely in a subsequent pregnancy, potentially leading to PTB.

A historical cohort study in 2015 of 878 women from the US identified that full dilatation CS increased the risk of spontaneous PTB (sPTB) in a subsequent pregnancy in comparison to CS in the first stage of labour [7]. Subsequently, in 2017 a large Canadian cohort study found a significantly increased risk of PTB in women delivered by second stage CS than those who delivered vaginally; supporting the hypothesis that full dilatation CS increases the risk of sPTB [8]. In 2019, the UK Preterm Clinical Network produced commissioning guidance to establish national pathways of care for women at risk of PTB [9]. The UK network identified women with a previous delivery by CS at full dilatation as being at intermediate risk of PTB, based on the above studies, and therefore recommend surveillance in pregnancy in a preterm prevention clinic. However, the aforementioned studies have not accounted for factors known to increase the risk of PTB, such as short interpregnancy interval [10] or previous cervical treatments [11].

The aim of this study was to determine if full dilatation CS increased the risk of sPTB in a subsequent pregnancy in comparison to CS in the first stage of labour in our unit, accounting for confounding factors known to increase the risk of sPTB. Furthermore, we wanted to assess a dose response effect by determining if progressively increasing cervical dilatation at the time of CS progressively increased the risk of sPTB in a subsequent pregnancy.

## 2. Materials and Methods

The population of interest were women delivered by emergency CS ≥ 37 weeks’ gestation between 2008–2015 (‘the index pregnancy’), who had a subsequent pregnancy delivered ≥16 weeks’ gestation until the end of 2015 (‘the subsequent pregnancy’). Women were retrospectively identified from the University Hospital’s Coventry and Warwickshire local maternity system Evolution v4.0 from the Trusts Performance and Programme Management Office in 2016. Every delivery ≥23 weeks’ gestation is recorded in this system, immediately post-delivery. The local electronic system did not record late miscarriages 16–23 weeks’ gestation; hence, these were searched for in the clinical record. This meant handwritten clinical case notes and data inputted into the hospital electronic record system were then meticulously reviewed to assess for eligibility, exposure and outcomes. There were no changes to practice standards over the study period.

Women were excluded from the study if there was a history of previous PTB < 37 weeks’ gestation; no documentation of gestation of previous deliveries, no documentation of the dilatation prior to delivery or of the delivery itself; previous inclusion in the study; delivery of the subsequent pregnancy at a different NHS Trust; index pregnancy an elective CS, intrauterine death or multiple gestation and inability to obtain clinical notes. Women with a CS prior to the index pregnancy were eligible to be included. Women with a previous CS were included as these are at increased risk of extensions to the uterine incision because of scarring and thinning of the lower uterine segment following the initial CS. During the study period, no cervical length assessment was undertaken for women in the study group.

Eligible women were divided into three groups to assess dose response as the risk of uterine incision extension into the cervix at CS would increase with increasing cervical dilatation. The reference group was women 0–5 cm dilated at index CS. The exposure groups were women 6–9 cm and fully dilatated at the time of index CS.

The primary outcome was sPTB < 37 weeks’ gestation in the subsequent pregnancy.

Baseline demographics were collected for the index and subsequent pregnancy to assess for confounding. All handwritten medical notes, operation notes and electronic documentation was reviewed and inputted into an electronic spreadsheet. Data collected for the index pregnancy included: age at delivery (<18, 18–34, and ≥35), ethnicity (white, black, Asian, other), body mass index at booking (≤18.5 kg/m^2^, 18.6–24.9 kg/m^2^, 25.0–25.9 kg/m^2^, and ≥30 kg/m^2^), parity (primigravida or multigravida), history of previous pregnancy loss between 12–24 weeks’ gestation, history of previous CS, gestation of delivery at index pregnancy (37–38, 39–40 and 41–42), smoking status at booking, history of domestic violence, history of illicit drug use, history of large loop excision of the transformation zone (LLETZ) and the presence of essential hypertension, pregnancy-induced hypertension, pre-eclampsia and gestation diabetes mellitus in the index pregnancy. Further information was collected about the index pregnancy delivery. This included if the labour was induced, the duration of labour (defined as from 4 cm dilatated in the presence of regular uterine contractions until delivery of the foetal body subdivided into 0–5 h, 6–11 h, 12–17 h, 17–23 h or ≥24 h) and the indication for the index CS (divided into foetal distress, failure to progress in the 1st and 2nd stages of labour, failed instrumental, not cephalic, a combination of failure to progress and foetal distress and other which includes previous CS, sepsis and antepartum haemorrhage). Operative notes were reviewed in depth and the following variables were recorded: the estimated blood loss at CS (mls) (subdivided into <500 mls, 500–1500 mls and ≥1499 mls), if an extension was sustained at index CS, and the index neonate’s birthweight (kg) (subdivided into <2.5 kg, 2.5–3.9 kg and >4 kg). An extension at CS was defined as an involuntary tear to the uterine incision in a different direction than that intended by the surgeon. An intentional increase in the incision length performed with scissors or a knife by the surgeon was not classified as an extension as these are performed away from the cervix. For the subsequent delivery data was collected and recorded for the interpregnancy interval (defined as the day of delivery to the last menstrual period for the next pregnancy (years), subdivided into <1 and >1). The Royal College of Obstetricians and Gynaecologists Green-top Guideline No. 45 states that an inter-delivery interval of less than one year can increase the risk of uterine rupture [12]. Therefore, we defined a short interpregnancy interval as being less than one year, based upon the Royal College of Obstetrics and Gynaecology. Additionally, for the subsequent pregnancy data was collected and recorded for the gestation of delivery (classified into <28 weeks’, 28–32 weeks’, 32–37 weeks’ and ≥37 weeks’), indication for delivery if preterm (classified into spontaneous, preterm prelabour rupture of membranes, iatrogenic and other) and mode of delivery.

The sample size (*n* = 1459) was determined after identifying women who met the study inclusion criteria. A post hoc power calculation was then performed, prior to analysing the data. 22.6% of this cohort were delivered by second stage CS in the exposure group. The Levine et al. cohort study found the PTB rate in a subsequent pregnancy was 2.3% from women delivered in the first stage of labour and 13.5% in the second stage of labour in the index pregnancy [7]. These PTB rates were entered into power calculation, meaning that this study would have >90% power to detect a 11% difference in PTB rate between the reference and exposure groups with a 95% confidence limit.

Data analysis was conducted in two stages using SPSS Statistics Version 24 (IBM, SPSS Statistics, Armonk, NY, USA) and GraphPad Prism version 7.00 (GraphPad Software, La Jolle, CA, USA). Firstly, heterogeneity was assessed between the reference and exposure groups. Normality was assessed for continuous variables; if the data was normally distributed ANOVA was used to assess for heterogeneity. If the data were not normally distributed, Kruskal–Wallis one-way ANOVA on ranks was used. The chi-squared test or Fisher’s exact test was used for categorical variables, when applicable, to assess for heterogeneity. A *p* value > 0.05 assumed no heterogeneity between the reference and exposure groups.

Odds ratios (OR) with corresponding 95% confidence intervals (CI) were calculated for every variable. The variables with the largest statistically significant odds of increasing the risk of sPTB and those thought clinically significant (extension of the uterine incision in the index pregnancy) were incorporated into a binary logistic regression model. OR and *p* values were then calculated in the multivariable analysis. Statistical significance was set at *p* < 0.05.

Ethical approval was granted from the Governance Arrangements for Research and Ethics Committee at University Hospital’s Coventry and Warwickshire on the 26 April 2016 (GR0104). Further approval was granted from University of Warwick Biomedical and Scientific Research Ethics Committee on the 3 August 2017 (REGO-2017-2085). The study has been reported following the STROBE statement v4 [13].

## 3. Results

The initial search identified 5808 women who had an emergency CS between 2008–2015. Of these, 1459 (25.1%) went on to have a subsequent delivery until 2015. Figure 1 shows the study population.

6.2% of the cohort had a PTB in the subsequent pregnancy, of which 64.8% were spontaneous. The reference group consisted of 707 women (48.5%) who were 0–5 cm dilated during the index CS. 329 women (22.6%) were fully dilated and 423 women (29.0%) were 6–9 cm dilated during the index CS; these were the exposure groups.

The exposure and reference groups baseline characteristics were compared to assess for heterogeneity. This is summarised in Table 1. Further baseline characteristics can be found in the Appendix A
Appendix A. There were significant differences between the groups in ethnicity, parity, history of previous CS, gestation of delivery in index pregnancy, smoking status, presence of gestational diabetes in the index pregnancy, if index labour was induced, the duration of the index labour, the indication for index CS, the estimated blood loss at index CS, the presence of extension at index CS and the birthweight of the index neonate.

To assess for confounding odds ratios were calculated for all baseline variables believed to increase the odds of sPTB in a subsequent pregnancy (Appendix A). A history of gestational diabetes in the index pregnancy, estimated blood loss <500 mls at the index CS, the index pregnancy gestation of delivery between 37–38 weeks’ gestation, a birthweight of the index infant of <2.5 kg and an interpregnancy interval of less than one year were all found to statistically significantly increase the odds of sPTB in the subsequent pregnancy. Therefore, these were incorporated into the binary logistic regression model (Table 2). One hundred and forty-three (9.8%) women sustained an extension in the index CS. The sustainment of an extension at index CS did not significantly increase the odds of sPTB in a subsequent pregnancy in the unadjusted (OR 1.11; 95% CI 0.47–2.64; *p* 0.82) and adjusted analysis (OR 1.34; 95% CI 0.54–3.29; *p* 0.53) (Table 2). Even though this did not significantly increase the odds of sPTB in the subsequent pregnancy, this was included into the binary logistic regression modelling as it was clinically significant. In this cohort smoking, maternal age, history of domestic violence and history of drug use did not significantly increase the risk of sPTB in the subsequent pregnancy (Appendix A
Appendix A).

Fifty-nine women (4.0%) had a sPTB in their subsequent delivery. The sPTB rate in a subsequent pregnancy was 3.5%, 4.0% and 5.2% for the reference, 6–9 cm dilatated and fully dilated at index CS groups (*p* 0.46). In comparison to the reference group, being 6–9 cm or fully dilated at index CS did not significantly increase the risk of spontaneous PTB < 37 weeks’ gestation in a subsequent pregnancy OR 1.14 (95% CI 0.62–2.08; *p* 0.68) and OR 1.49 (95% CI 0.80–2.72; *p* 0.22) respectively in the unadjusted analysis (Table 3). After adjusting for potential confounders, the risk of sPTB in a subsequent pregnancy after an emergency CS increased but did not reach statistical significance (6–9 cm: OR 1.64; 95% CI 0.83–3.28; *p* 0.158 and full dilatation: OR 1.86; 95% CI 0.91–3.83; *p* 0.090).

There were no statistically significant differences between the exposure and reference groups in the odds of sPTB at <28 and <32 weeks’ gestation (Table 3).

To further assess the risk of increasing cervical dilatation at emergency CS on sPTB in a subsequent pregnancy, binary logistic regression modelling was undertaken. For each 1 cm of cervical dilatation gained at the index CS the odds of subsequent sPTB in a subsequent pregnancy increased by 2%, but the relationship did not reach statistical significance (OR 1.02; 95% CI 0.94–1.11; *p* 0.65). After adjusting for confounding the odds of sPTB for each centimetre of cervical dilatation increased but did not reach statistical significance (aOR 1.08; 95% CI 0.99–1.18; *p* 0.11).

A surprise finding was that despite the fact that in the UK women are advised not to conceive again within a year after a CS, there was an increased risk of sPTB following an emergency CS with a short interpregnancy interval. Therefore, a secondary analysis on the effect of short interpregnancy interval following an emergency CS was undertaken. The adjusted OR for sPTB after an emergency CS with an interpregnancy interval <1 year was 3.01 (95% CI 1.72–5.26; *p* < 0.000). For each 6 months increment in interpregnancy interval the odds of sPTB decreased; however, this did not reach statistical significance in the unadjusted and adjusted analysis (OR 0.90; 95% CI 0.81–1.01; *p* 0.75 and aOR 0.91; 95% CI 0.80–1.02; *p* 0.11, respectively).

## 4. Discussion

This study’s aim was to determine if full dilatation CS increased the risk of sPTB in a subsequent pregnancy in comparison to CS in the first stage of labour in our unit, while accounting for confounding factors known to increase the risk of sPTB. Furthermore, we wanted to determine if increasing cervical dilatation at the time of CS increased the risk of sPTB in a subsequent pregnancy.

Full dilatation CS has been associated with increased risk of sPTB in a subsequent pregnancy. During full dilatation CS there is an increased risk of extension of the hysterotomy incision into the cervix [4,5]. Furthermore, when the cervix is incorporated into the lower segment, a low hysterotomy incision could inadvertently cross the cervix, or the cervix could be brought into the hysterotomy closure. This could lead to the loss of the cervix’s structural integrity making it weak in a future pregnancy. The concept of surgery leading to cervical weakness is supported by the evidence that previous cervical treatments increase the risk of PTB in subsequent pregnancies, secondary to cervical weakness [11,14].

We found that CS performed at 6–9 cm dilated and fully dilated did increase the risk of sPTB < 37 weeks’ gestation in a subsequent pregnancy, however the relationship did not reach statistical significance. CS at 6–9 cm and full cervical dilatation did not increase the risk of sPTB < 28 and <32 weeks’ gestation in a subsequent pregnancy. Therefore, our hypothesis that increasing cervical dilatation at CS increases the risk of sPTB in a subsequent pregnancy was not supported.

Our results differ from other studies that have been published which do show a significant association between full dilatation CS and subsequent PTB. In 2015 Levine et al. found 6-fold higher odds of sPTB when CS was performed in the second stage in comparison to the first stage of labour, a relationship which remained after adjusting for race, chronic hypertension, induction of labour, but not short interpregnancy interval [7]. Following this a large Canadian retrospective cohort study of 189,021 women found an increased risk of sPTB < 37 weeks’ gestation in a subsequent pregnancy when CS was performed in the second stage of labour in comparison to vaginal delivery [8]. The risk of subsequent delivery <32 weeks’ gestation was greater in their analysis, that accounted for body mass index and smoking but not short interpregnancy interval. A third large retrospective cohort study of 2675 women from Australia found that full dilatation CS doubled the odds of sPTB in a subsequent pregnancy in comparison to CS in the first stage of labour [15].

The current study found a non-statistically significant increase in the risk of sPTB in a subsequent pregnancy after a full dilatation CS in comparison to a CS in the first stage of labour. The current study was powered to detect a difference in the sPTB rate of 2.3–13.5% between first and second stage CS but found a difference of 2.3–5.3%. Therefore, the study was underpowered to detect a small increase in PTB with full dilatation CS. Our results may also differ to other published results as we have adjusted for short interpregnancy interval. In this study, the logistic regression analysis found short inter-pregnancy interval of <1 year increased the risk of sPTB more than full dilatation at index CS or extension of uterine incision. This may account for the differences seen in our study to other published results. Furthermore, we have included multiparous women in the analysis, as did the Levine et al. cohort [7], whereas the other cohorts have included nulliparous women [8,15]. A total of 17.6% of this cohort were multiparous, and 0.07% of women in the cohort had a CS prior to the index pregnancy. A weakness of this study is that we do not know the dilatation of the CS prior to the index pregnancy, which could confound results. However, the numbers of women with a previous CS were low and we did not find that CS prior to the index pregnancy increased the risk of sPTB in the follow up pregnancy.

Additionally, we found in the logistic regression analysis that index CS performed at 37–38 weeks’ gestation significantly was associated with an increased the risk of sPTB in a subsequent pregnancy in comparison to CS at 39–42 weeks’ gestation. Even though women with an elective CS were excluded from this study, this finding may be important with regards to timing of elective procedures and the risk on future pregnancies. However, it is difficult to draw conclusions around this as the study was not powered to look at time of delivery and risk of sPTB in a subsequent pregnancy and time of delivery should be carefully balances upon the risks posed to that pregnancy. In this population, an estimated blood loss of <500 mls at the index CS was associated with an increased risk of sPTB in the subsequent pregnancy. It is difficult to ascertain the clinical significance of this finding as blood loss at CS is often inaccurate and this finding could be due to chance. We have theorised that these women may have increased myometrial sensitivity to agonists, leading to an improved tonic contraction of the uterus to prevent haemorrhage and earlier activation of parturition pathways [16].

The major strength of this study was the comprehensive data collection, vigorous exclusion criteria, and categorisation of the variables. For all women included in the study, clinical notes and operative notes were reviewed thoroughly in detail to determine the reliability of the data and to assess for all variables that could confound PTB rates. Furthermore, the data all originated from one hospital trust, therefore decreasing variation in clinical practice across units.

The first limitation was described above, detailing how this study was underpowered to detect a small difference in outcome. Another limitation is that the electronic hospital information system used is only able to capture deliveries >22 weeks’ gestation. Women with a late miscarriage would not have been captured in the analysis. A case control study from 2017 found that CS at full dilatation increased the risk of late miscarriage and sPTB [17]. The increase in late miscarriages would not have been detected.

A surprise finding from this study was that a short interpregnancy interval of less than 1 year had a greater impact upon the risk of sPTB in a subsequent pregnancy than full dilatation CS in the logistic regression analysis. Previous work has also demonstrated that a short interpregnancy interval increases the risk of sPTB in a subsequent pregnancy [10]. This has been hypothesised as being secondary to maternal nutritional depletion and postpartum stress [18]. However, further work is required to determine if the sPTB rate differs for women with a short interpregnancy interval who were delivered vaginally or by CS.

The current data adds to the emerging literature associating previous CS with sPTB. In 2019 the UK Preterm Clinical Network produced commissioning guidance to establish national pathways of care for women at risk of PTB [9]. They have identified women with a previous delivery by CS at full dilatation as being at intermediate risk of PTB and therefore recommend surveillance in pregnancy in a preterm prevention clinic. Further to this in 2019 NHS England released the ‘Saving Babies’ Lives Version Two’ Care Bundle in which women with a history of full dilatation CS was also deemed being an intermediate risk for the development of PTB [19]. Based on the published literature and national care pathways, we would recommend that women with a previous full dilatation CS are referred for increased surveillance in their pregnancy. However, we also recommend that women with a short interpregnancy interval after a CS should also be considered for referral to preterm prevention services to decrease the risk of sPTB. We would advise emphasis on postnatal contraception after caesarean to prevent pregnancies with a short interpregnancy interval.

## 5. Conclusions

The current study of our population found that full dilatation CS did not significantly increase the risk of sPTB in a subsequent pregnancy. We found that a short interpregnancy interval following CS increased the risk of sPTB more than a full dilatation CS. Clinically, this is important as improving counselling about postnatal contraception following a CS and increased surveillance in women with a short interpregnancy interval may be effective strategies to reduce PTB. Further prospective studies are required to confirm our findings in other populations and are current recruiting [20]. 

## Figures and Tables

**Figure 1 jcm-09-03998-f001:**
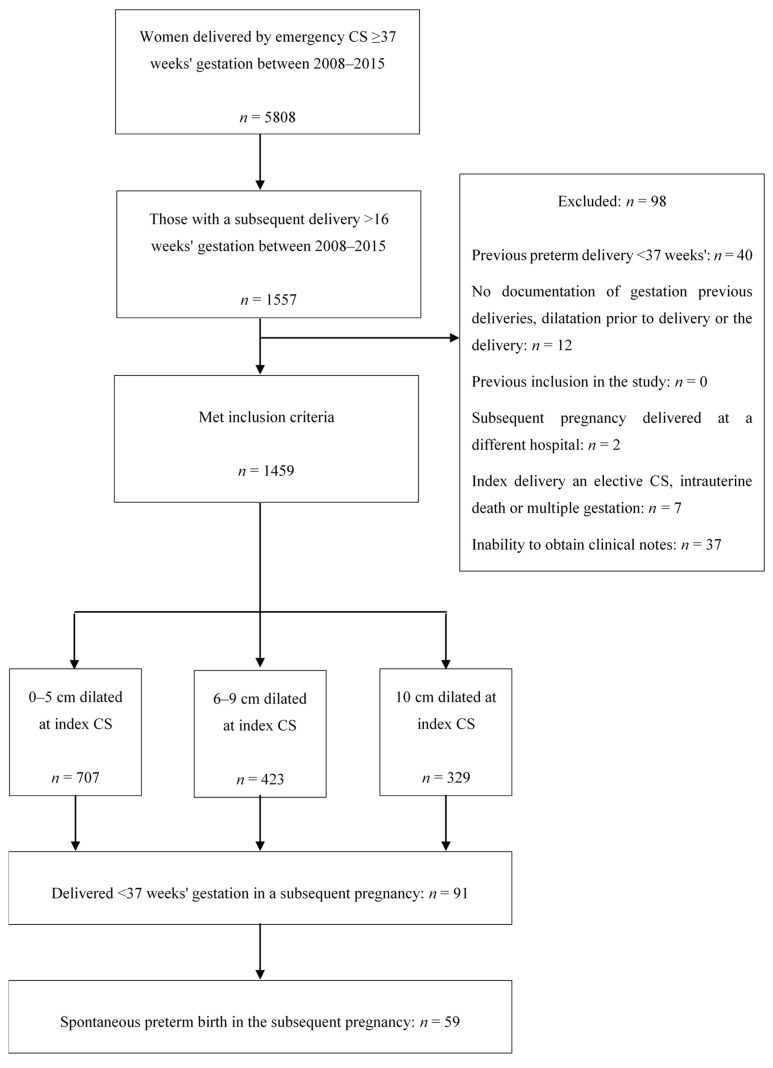
A flow diagram of the study population. CS: Caesarean section.

**Table 1 jcm-09-03998-t001:** Baseline characteristics of the study population.

		Cervical Dilatation at Index Caesarean Section			
		0–5 cm (*n* = 707)	6–9 cm (*n* = 423)	Full (*n* = 329)	*p*
		*n* (%)	*n* (%)	*n* (%)	
Age (years)	<18	11 (1.6)	5 (1.2)	5 (1.5)	0.79
	18–34	634 (89.7)	389 (92.0)	301 (91.5)	
	≥35	62 (8.8)	29 (6.9)	23 (7.0)	
Ethnicity	White	423 (59.8)	286 (67.6)	244 (74.2)	<0.001
	Black	120 (17.0)	57 (13.5)	21 (6.4)	
	Asian	140 (19.8)	59 (13.9)	49 (14.9)	
	Other	24 (3.4)	21 (5.0)	15 (4.6)	
Body mass index (kg/m^2^)	≤18.5	17 (2.4)	9 (2.1)	7 (2.1)	0.19
	18.6–24.9	315 (44.6)	209 (49.9)	167 (50.8)	
	25.0–29.9	203 (28.8)	129 (30.5)	93 (28.3)	
	≥30.0	171 (24.2)	76 (18.0)	62 (18.8)	
Parity	Primigravida	555 (78.5)	351 (83.0)	305 (92.7)	<0.001
	Multiparous	152 (21.5)	72 (17.0)	24 (7.3)	
Previous caesarean section	Yes	67 (9.5)	22 (5.2)	7 (2.1)	<0.001
	No	640 (90.5)	401 (94.8)	322 (97.9)	
Gestation index delivery (weeks’)	37–38	151 (21.4)	50 (11.8)	47 (14.3)	0.007
	39–40	287 (40.6)	199 (47.0)	150 (45.6)	
	≥41	269 (38.0)	174 (41.1)	132 (40.1)	
Smoking status	Current smoker	125 (17.7)	44 (10.4)	46 (14.0)	0.003
	Non-smoker	582 (82.3)	379 (89.6)	283 (86.0)	
Gestational diabetes in the index pregnancy	Yes	42 (5.9)	11 (2.6)	9 (2.7)	0.01
	No	665 (94.1)	412 (97.4)	320 (97.3)	
Induction of labour in index pregnancy	Yes	441 (62.4)	201 (47.5)	133 (40.4)	<0.001
	No	266 (37.6)	222 (52.5)	196 (59.6)	
Indication for index pregnancy caesarean section	Fetal distress	374 (52.9)	142 (33.6)	60 (18.2)	<0.001
	Failure to progress 1st stage	155 (21.9)	204 (48.2)	0 (0.0)	
	Failure to progress 2nd stage	0 (0.0)	0 (0.0)	163 (49.5)	
	Failed instrumental	0 (0.0)	0 (0.0)	78 (23.7)	
	Not cephalic	73 (10.3)	26 (6.1)	8 (2.4)	
	Failure to progress and fetal distress	45 (6.4)	38 (9.0)	14 (4.3)	
	Other	60 (8.5)	13 (3.1)	6 (1.8)	
Extension at index caesarean section	Yes	42 (6.2)	49 (12.4)	52 (16.8)	<0.001
	No	632 (93.8)	347 (87.6)	258 (83.2)	
Interpregnancy interval (years)	<1	114 (16.1)	63 (14.9)	47 (14.3)	0.71
	>1	593 (83.9)	360 (85.1)	282 (85.7)	

**Table 2 jcm-09-03998-t002:** Odds ratio for all baseline variables found to increase the risk of spontaneous preterm birth in a subsequent pregnancy. sPTB: spontaneous preterm birth; CS: caesarean section; OR: odds ratio; aOR: adjusted odds ratio.

		sPTB < 37 Weeks’ Gestation	Delivery ≥ 37 Weeks’ Gestation	OR (95% CI)	*p*	aOR (95% CI)	*p*
		(*n*)	(*n*)				
Gestation delivery index pregnancy	39–42	37	1174	1.00		-	
	37–38	22	226	3.09 (1.79–5.34)	<0.001	2.61 (1.40–4.87)	0.003
Gestational diabetes in the index pregnancy	No	53	1344	1.00		-	
	Yes	6	56	2.72 * (1.20–6.40)	0.04	2.38 (0.88–6.45)	0.089
Estimated blood loss at index CS (mls)	≥500	29	956	1.00		-	
	<500	24	372	2.13 (1.22–3.70)	0.007	2.05 (1.14–3.67)	0.016
Extension at index CS	No	47	1190	1.00		-	
	Yes	6	137	1.11 (0.47–2.64)	0.82	1.34 (0.54–3.29)	0.53
Birthweight index neonate (kg)	≥2.5	52	1353	1.00		-	
	<2.5	7	47	3.88 (1.67–8.99)	<0.001	2.24 (0.82–6.13)	0.115
Interpregnancy interval (years)	>1	39	1196	1.00		-	
	<1	20	204	3.01 (1.72–5.26)	<0.001	3.10 (1.71–5.61)	<0.001

* Fishers Exact test.

**Table 3 jcm-09-03998-t003:** Risk of spontaneous preterm birth in a subsequent pregnancy at <28, <32 and <37 weeks’ gestation following an emergency caesarean section at 0–5, 6–9 of full cervical dilatation.

		*n* (%)	OR (95% CI)	*p*	aOR (95%CI)	*p*
<28 weeks’ gestation	0–5 cm dilated at index CS	3 (0.42)	1.00 (ref)		1.00 (ref)	
	6–9 cm dilated at CS	3 (0.71)	1.68* (0.34–8.35)	0.68	1.81 (0.33–10.01)	0.5
	Fully dilated at index CS	2 (0.61)	1.44* (0.25–8.64)	0.66	0.78 (0.07–8.11)	0.83
<32 weeks’ gestation	0–5 cm dilated at index CS	7 (0.99)	1.00 (Ref)		1.00 (ref)	
	6–9 cm dilated at CS	4 (0.95)	0.95 * (0.28–3.28)	>0.99	1.11 (0.31–4.00)	0.87
	Fully dilated at index CS	5 (1.52)	1.54* (0.49–4.90)	0.53	1.40 (0.39–5.03)	0.61
<37 weeks’ gestation	0–5 cm dilated at index CS	25 (3.54)	1.00 (ref)		1.00 (ref)	
	6–9 cm dilated at CS	17 (4.02)	1.14 (0.62–2.08)	0.68	1.64 (0.83–3.28)	0.16
	Fully dilated at index CS	17 (5.17)	1.49 (0.80–2.72)	0.22	1.86 (0.91–3.83)	0.09

* Fishers Exact test.

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
