# Peer review of "Full Dilatation Caesarean Section and the Risk of Preterm Delivery in a Subsequent Pregnancy: A Historical Cohort Study"

_jcm, 2020, doi:10.3390/jcm9123998_

Round 1

Reviewer 1 Report

The purpose of this study is to determine whether full dilatation cesarean delivery increases the risk of spontaneous preterm birth (sPTB) < 37 weeks in a subsequent pregnancy compared to cesarean deliveries done during the first stage of labor. This was a retrospective cohort study of women who had cesarean delivery during a 7 year period (from 2008 to 2015) in a single University Hospital. They identified 1,557 women who met inclusion criteria. There were no statistically significant differences in the preterm birth rate between groups. Ideally they should have done a power calculation before data mining to answer their research question or primary outcome. They did a post hoc analysis that reported > 90% power to detect an 11% difference in sPTB. It is not clear to me what they did to determine the > 90% power and I recommend this manuscript to undergo review by a statistician. They do not indicate that a secondary outcome of the study was to evaluate the effect of a short interpregnancy interval (SIPI), which was defined as < 1 year or < 12 months, on the rate of sPTB. There is also no calculated sample size done during the planning phase to answer this research question. They report that a SIPI < 1 year significantly increased the odds of of sPTB in the subsequent pregnancy. They hypothesized “that time of healing may be more important than the extent of damage”. The authors acknowledge that there is a known association between a SIPI and an increase sPTB rate. Not a new finding.  Of note, there is no standard definition for SIPI. It has been defined as less than 3, 6, 9, 12, or 18 months. The authors only found a statistically significant increase in sPTB using a definition of < 12 months; which suggest a chance phenomenon. Are the authors saying that SIPI < 12 months after cesarean deliveries increases the odds of having sPTB in subsequent pregnancies but not after vaginal births? I believe they would need to compare the sPTD rate between women with SIPI who had vaginal and cesarean deliveries. I would recommend for the authors to accept the non-significant differences in their study. They should focus the discussion comparing their study results with the published literature. They erroneously state that “in contrast to other studies, we excluded women with PTB”.  A study by Levine et al (Am J Ob Gyn 2015;212: 360.e1-7) excluded women who were preterm in their index pregnancies or those with a history of PTB at the time of their index pregnancy. There is merit to the publication of negative results.

Reviewer 2 Report

This is an interesting study that adds to the literature in this area.

Questions to the authors:

  1. Were there any changes to practice standards over the period of study that may impact the results?
  2. It has been previously documented that short interpregnancy interval is a risk for preterm delivery and small for gestational age babies possibly due to maternal age or lack of opportunity for the mother to replenish necessary nutritional health. (Association of short interpregnancy interval with pregnancy outcomes according to maternal age JAMA Dec 2018 Schummers et al; J Perinatol. Sep 2019 39(9) 1175-1181). Did your group look at other factors that may impact preterm birth - nutrition/maternal age/smoking/drug use/socioeconomic factors etc.?
  3. Was any assessment done of cervical length or anatomy by ultrasound in this group?
  4. Given that this study has conflicting data to other studies, but was underpowered, how to you suggest this would impact patient counselling or practice patterns?

Reviewer 3 Report

This is a clearly written manuscript and much work went into collecting all the data. I have a few concerns that should be addressed before the paper is further considered for publication.

In contrast to other studies, in this work cervical dilation during the index Cesarean delivery was found to have no significant effect on the risk of sPTB in the subsequent pregnancy. But pregnancies that may have preceded the index CS are not considered. Is this how other similar studies cited were designed? If, for example, the index CS was performed in the absence of cervical dilation, but there was a CS before that performed in the presence of dilation, could that confound the results? This problem could be addressed by taking previous CS deliveries into account when analyzing the data or eliminating cases where previous CS deliveries preceded the index CS delivery.

In addition to highlighting short pregnancy interval as a risk factor for sPTB, gestational age at the time of the index CS, which also had a significant effect on sPTB risk in the subsequent pregnancy, should be highlighted. Although elective CS deliveries were not included in this study, this finding may still have implications for the timing of elective procedures.

Why does less blood loss in the index CS correlate with increased risk of sPTB in the subsequent pregnancy? Please discuss this counterintuitive finding.

P does not truly equal 0 in those instances where it is so stated. P should be expressed as <.001 in those instances.

Finally, there are minor grammatical errors throughout the manuscript. For example, line 49 is not a complete sentence and "weeks gestation" should be "weeks' gestation."

Round 2

Reviewer 1 Report

Full dilatation cesarean manuscript revision review:

This revised manuscript requires further revisions.

This manuscript does not suggest “that time for healing may be more important than the extent of damage at CS” since vaginal births with short interpregnancy intervals (SIPI) have been associated with sPTB in subsequent pregnancies. How is that statement applicable to vaginal births? I recommend removing that statement from the abstract and discussion sections; and just stating their findings. I would recommend that the authors embrace the currently accepted hypotheses that explain the association between SIPI and sPTB (maternal depletion and postpartum stress).

This study, from the start, was not planned to have an adequate power to detect the hypothesized effect. The paragraph on page 4 line 124 regarding power analysis is poorly written and incorrect. The study sample size was 1, 459 (met inclusion criteria) as shown in Figure 1 rather than 1,557. The second stage CS delivery group was 22.6% (329/1459) not 22.3%. It should be clearly stated that the sample size was determined or established after identifying the women who met the study inclusion criteria. Then a post hoc power calculation was done. I believe that most editors and reviewers would consider inappropriate doing post hoc power analysis from a negative study.

The authors should be aware that a statistical associations in a logistic regression analysis does not mean causation. The finding that an estimated blood loss of less than 500 ml was a risk factor for sPTD is most likely chance phenomenon with no clinical significance. The authors should be aware of the inaccuracy of estimated blood loss when compared to quantitative blood loss during parturition. The current trend is to abandon estimated blood loss and document quantitative blood loss during child birth. Additionally, the average blood loss at CS is approximately 800 ml. Very hard to believe having a blood loss less than 500 ml during an emergency CS delivery. Most surgeons are known to underestimate blood loss. The authors should remove their explanation as to why less blood loss is associated with sPTB from the manuscript unless they have references (either animal or basic science data) that support their statement.  I do not believe their explanation.  

My understanding is that the linear regression analysis identified index CS performed at 37 - 38 weeks, blood loss less than 500 ml, and SIPI less than 1 year as being associated with an increased risk of sPTB. Please note that the finding of statistical associations in a logistic regression analysis does not mean causation. The authors must interpret the plausible clinical importance or reasonable applications of their data using common sense.  All three are associations that must be taken with a grain of salt.  

Reviewer 3 Report

All of my concerns have now been adequately addressed.

There are several grammatical errors throughout the manuscript, e.g. the correct phrase is "weeks' gestation," not "week's gestation."
